# Balance Rehabilitation with Peripheral Visual Stimulation in Patients with Panic Disorder and Agoraphobia: An Open-Pilot Intervention Study

**Daniela Caldirola** [1,2,3], **Claudia Carminati** [2,3], **Silvia Daccò** [1,2,3], **Massimiliano Grassi** [1,2], **Giampaolo Perna** [1,2,3] **and Roberto Teggi** [4,*]

1 Department of Biomedical Sciences, Humanitas University, Via Rita Levi Montalcini 4, 20090 Pieve Emanuele, Italy; daniela.caldirola@hunimed.eu (D.C.); silvia.dacco@hunimed.eu (S.D.); massimiliano.grassi@hunimed.eu (M.G.); giampaolo.perna@hunimed.eu (G.P.)
2 Department of Clinical Neurosciences, Villa San Benedetto Menni Hospital, Hermanas Hospitalarias, Via Roma 16, 22032 Albese con Cassano, Italy; carminaticlaudia@gmail.com
3 Humanitas San Pio X, Personalized Medicine Center for Anxiety and Panic Disorders, Via Francesco Nava 31, 20159 Milan, Italy
4 Department of Otolaryngology, San Raffaele Scientific Hospital, Via Olgettina 60, 20132 Milan, Italy
* Correspondence: teggi.roberto@hsr.it

**Abstract:** Given the involvement of balance system abnormalities in the pathophysiology of panic disorder and agoraphobia (PD-AG), we evaluated initial evidence for feasibility, acceptability, and potential clinical usefulness of 10 sessions of balance rehabilitation with peripheral visual stimulation (BR-PVS) in an open-pilot 5-week intervention study including six outpatients with PD-AG who presented residual agoraphobia after selective serotonin reuptake inhibitor (SSRI) treatment and cognitive–behavioral therapy, dizziness in daily life, and peripheral visual hypersensitivity measured by posturography. Before and after BR-PVS, patients underwent posturography, otovestibular examination (no patients presented peripheral vestibular abnormalities), and panic-agoraphobic symptom and dizziness evaluation with psychometric tools. After BR-PVS, four patients achieved postural control normalization measured by posturography, and one patient exhibited a favorable trend of improvement. Overall, panic-agoraphobic symptoms and dizziness decreased, even though to a lesser extent in one patient who had not completed the rehabilitation sessions. The study presented reasonable levels of feasibility and acceptability. These findings suggest that balance evaluation should be considered in patients with PD-AGO presenting residual agoraphobia and that BR-PVS might be an adjunctive therapeutic option worth being tested in larger randomized controlled studies.

**Keywords:** panic disorder; agoraphobia; balance; rehabilitation; visual; dizziness

## 1. Introduction

Clinical and experimental evidence has suggested the involvement of balance system abnormalities in the pathophysiology of panic disorder, especially with comorbid agoraphobia (PD-AG). Around 50–70% of patients with panic disorder complain of dizziness or unsteadiness both during and outside panic attacks (PAs) [1,2]. Preliminary associations between dizziness during PAs and agoraphobia have been found, and observations in clinical practice have suggested a link between dizziness outside PAs and the presence and severity of agoraphobia [3].

Otoneurologic examinations have revealed a higher prevalence of peripheral vestibular abnormalities in patients with PD-AG [4–8]. Even when full-blown vestibular abnormalities are absent, these patients often present impaired inter-sensory integration in balance control during posturography and unusual dependence on information from non-vestibular sensory channels, primarily visual. Consequently, they exhibit reduced adaptability to postural

challenges and misleading sensory clues, with subjective intolerance and objective impairment in balance-challenging situations such as in moving visual environments [3,9–14]. In line with these results, we found postural hypersensitivity to visual stimuli moving in the peripheral visual field during posturography in patients with PD-AG compared with healthy controls [15]. Finally, dizziness and unsteadiness in patients with PD-G may belong to the spectrum of persistent postural–perceptual dizziness (PPPD), possibly sharing some common mechanisms [16,17].

Unfortunately, despite the possibility that some patients with panic disorder may have a "balance vulnerability" to agoraphobia [18,19], balance system evaluation is not part of routine clinical practice, even when persistent agoraphobia or complaints about dizziness and/or subjective visual hypersensitivity in daily life are present. Unrecognized balance system abnormalities might explain part of the unsatisfactory therapeutic responses in patients with PD-AG [20], and at least a portion of these patients might benefit from interventions to regain the inter-sensory integration in balance control.

Only one preliminary open study has explored this potential therapeutic option by applying vestibular rehabilitation in patients with PD-AG and various balance system abnormalities. Patients obtained an additional improvement in panic-agoraphobic symptoms to the partial benefits previously achieved with a 4-week behavioral therapy phase [21].

Considering our previous findings of postural hypersensitivity to peripheral visual stimulation in patients with PD-AG [15], this open-pilot intervention study aimed to evaluate initial evidence for feasibility, acceptability, and potential clinical usefulness of a newly developed balance rehabilitation with peripheral visual stimulation in patients with PD-AG who presented residual agoraphobia after recommended treatments and a balance system-related clinical profile. Throughout the manuscript, reporting and related discussion follow the current guidelines and recommendations for reporting non-randomized pilot and feasibility studies [22–24].

## 2. Materials and Methods

### 2.1. Study Design and Participants

This 5-week open-pilot intervention study included a sample of 6 participants (3 females and 3 males) with DSM-IV-TR-defined PD-AG [25], who were consecutively recruited among outpatients referring to a single senior psychiatrist (D.C.), which delivered psychiatric visits for 24 h per month, at the Anxiety Disorders Clinical and Research Unit of San Raffaele Turro Hospital, Milan. The entire study lasted from the beginning of March to the end of October 2009. Due to time and personnel constraints, the last possible enrollment was scheduled at the beginning of October to conclude the 3 week-balance rehabilitation by the end of the same month. Finally, due to organizational issues, enrollment was temporarily suspended during August. Hence, enrollment was conducted in 6 months.

A senior psychiatrist (DC) and a senior otorhinolaryngologist (RT) recruited participants assessing their eligibility via clinical interviews and static posturography (SP) with peripheral visual stimulation (PVS) (SP-PVS) according to the inclusion and exclusion criteria listed below. During clinical interviews by a senior psychiatrist (D.C.), psychiatric diagnoses were assessed through a DSM-IV-TR-based unstructured interview, and information concerning medical history and medications was collected.

In addition to the current diagnosis of PD-AG, inclusion criteria were the following: age $\geq$ 18 years; receiving recommended first-line antipanic medications [26] for at least 3 months, with adequate and stable dosages for at least 3 weeks before the beginning of the study, and having completed an adequate cognitive–behavioral therapy (CBT) program of at least 12 sessions, delivered by a licensed psychologist, according to guidelines for PD-AG [26]; being bothered by persistent residual agoraphobia despite appropriate treatments; presenting a balance system-related clinical profile including (1) having been bothered by at least 15 dizzy days per month in the last 3 months, (2) a Visual-Romberg Quotient (Visual-RQ) < 0.77 (see below) during the first static posturography with peripheral visual stimulation (SP-PVS) one week before the first rehabilitation session, indicating the pres-

ence of hypersensitivity to peripheral visual stimulation, and (3) a Visual-RQ < 0.77 also during the second SP-PVS just before the first rehabilitation session, as confirmation of the peripheral visual hypersensitivity; finally, voluntarily provided written informed consent to participate.

Patients were excluded in case of current co-morbidity with other psychiatric disorders; lifetime psychotic disorders; drug or alcohol abuse or dependence within the past 6 months; assumption of medications not recommended for PD-AG; assumption of benzodiazepines on a regular daily basis; lifetime neurological disorders, except for migraine; and pregnancy.

No changes in the type or dosages of antipanic medications were allowed during the study, while the occasional use of benzodiazepines, no more than twice a week, was allowed. Likewise, no CBT or other psychological interventions were allowed.

The study was in accordance with the Declaration of Helsinki and approved by the Local Health Authority (ASL) of Milan Ethics Committee ASL "City of Milan".

### 2.2. Procedure and Measures

At the beginning (T0, a week before the first session) and end (T1, a week after the last session) of rehabilitation, participants underwent comprehensive instrumental and clinical assessment, as follows. The entire assessment was conducted at the outpatient facilities of the Anxiety Disorders Clinical and Research Unit of San Raffaele Turro Hospital, Milan.

2.2.1. Balance System Function Evaluation

Conducted by a senior otorhinolaryngologist (RT), including SP-PVS [15], otovestibular evaluation with videonystagmography (Interacoustics–Assen–Denmark), and collection of medical history on vestibular disorders (only at T0). An additional SP-PVS with the same methodology was performed just before the first rehabilitation session to confirm the reliability of the visual hypersensitivity of each participant.

Patients had to refrain from alcohol and benzodiazepine use for at least 2 days, xanthines for at least 8 h, and food or smoking for at least 2 h before posturography-PVS [27].

Static posturography usually measures postural sway as the displacement of the center of pressure (CoP) of an individual when he/she stands on a platform with pressure transducers recording oscillations under different sensorial conditions. It provides information on the individual's ability to integrate multiple inputs in the control of posture [28].

For the aim of this study, we specifically evaluated peripheral visual hypersensitivity by SP-PVS, which encompassed three sequential conditions: open-eyes pre-visual stimulation, moving visual stimulation in the peripheral visual field, and open-eyes post-visual stimulation [15]. We used a force platform (Amplaid SveP, 10 Hz-signal acquisition) that conformed to the standards of the International Society of Posturology [29]. The recording in each condition took 30 s. Participants were instructed to stare at a white vertical line (20 cm long by 4 cm wide) on a black background projected on a 17-in. display in front of them without moving their heads or gazes during all three conditions. Peripheral visual stimulation was produced by a video-film (32 times-accelerated), showing people moving in various contexts of everyday life, and projected on two lateral 150-cm$^2$ screens; each screen formed a 30° angle with the line connecting the head of subjects and the central display and covering an angle from 10° to 50° of the visual field (Figure 1). Different moving scenes were used during the first SP-PVS (T0), the second one (just before the first rehabilitation session), and the final one (T1). In each progressive SP-PVS, the moving scenes were the same for all patients.

The same apparatus was used to deliver the balance rehabilitation (BR) with peripheral visual stimulation (BR-PVS) described in Section 2.3.

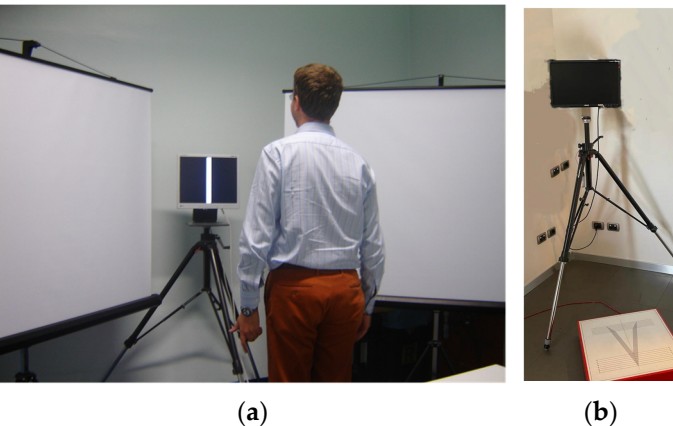

(**a**)                    (**b**)

**Figure 1.** Set-up of the apparatus used to perform the SP-PVS and deliver the BR-PVS. A participant and the lateral screens (**a**); the force platform (**b**) on which participants were standing during the procedures.

The posturographic outcome of interest in this study was peripheral visual hypersensitivity as identified by the Visual-RQ, a modified version of the Romberg Quotient (RQ) [27]. The Visual-RQ was the ratio between the length of body sway (i.e., the sum of the distances between the sequential sampled positions of CoP) [28] during the open-eyes-condition pre-peripheral visual stimulation and that during the open-eyes-condition with peripheral visual stimulation. The lower the Visual-RQ, the greater the postural instability induced by visual stimulation; the Visual-RQ cut-off value to identify the visual hypersensitivity was <0.77 [27].

Visual analog scales (VASs) for anxiety (VAS-A) and dizziness (VAS-D) (continuum from 0, no anxiety/dizziness, to 100, the worst anxiety/dizziness imaginable, on a horizontal line 100 mm in length) [30] were administered immediately before and after SP-PVS; the VASs-after included those referring to the visual stimulation and to the moment immediately after the end of SP-PVS.

The otovestibular evaluation included bedside head impulse test, head-shaking test, positional testing, research of skull vibration nystagmus, and Unterberger test; maneuvres to rule out benign paroxysmal positional vertigo were also performed.

### 2.2.2. Panic-Agoraphobic Symptom and Dizziness Evaluation

Conducted by a senior psychiatrist (D.C.) and a licensed psychologist (C.C.), including the Panic-Associated Symptom Scale (PASS), Mobility Inventory for Agoraphobia (MIA), and Dizziness Handicap Inventory (DHI), which are validated tools widely used for clinical and research purposes.

The PASS is a clinician-administered psychometric tool to quantify the frequency and intensity of unexpected and expected PAs and levels of anticipatory anxiety and AG during the last week. It provides a total score of 0 to 28 and subscale scores, assessing unexpected PAs (scores between 0 and 6), expected PAs (scores between 0 and 6), limited symptom PAs (scores between 0 and 4), anticipatory anxiety (scores between 0 and 7), and agoraphobia (scores between 0 and 5). The PASS has been widely used to measure clinical improvement during treatments [31].

The MIA is a self-administered questionnaire that evaluates the agoraphobic avoidance of different daily life situations. It includes two scales, namely the Avoidance Accompanied by a trusted Companion (AAC) scale and the Avoidance Alone (AAL) scale (26 and 27 items on a Likert-type scale, respectively; for each item, a rating of 1 indicates the situation was never avoided, and 5, always avoided). The final score of each scale is obtained by dividing the sum of all items by the number of items, producing scores between 1 and 5 [32,33].

The DHI is a self-administered 25-item questionnaire that assesses the self-perceived handicapping effects caused by dizziness in functional, emotional, and physical domains

of daily life, producing a total score of 0 to 100. Scores between 0 and 29 indicate a mild dizziness-related handicap, 30–60, moderate handicap, and >60 severe handicap [34,35]. As additional information, we evaluated the perception of dizziness during agoraphobic situations via an ad hoc modified version of the MIA (MIA-Dizziness, MIA-D). In the MIA-D, the 27 items of the MIA-AAL are listed; participants rate each item as "0" (never), "2" (sometimes), or "4" (always) to indicate how much they usually perceived dizziness when exposed to each situation, producing a total score ranging from 0 to 108.

For compilation of MIA, DHI, and MIA-D, participants were asked to refer to the last two weeks.

### 2.3. Balance Rehabilitation with Peripheral Visual Stimulation

Patients received 10 sessions of BR-PVS on an individual basis over 3 weeks (i.e., 3 sessions per week during the first 2 weeks and 4 sessions during the last week). Each session lasted 45 min. Rehabilitation was conducted by a senior otorhinolaryngologist (RT).

During each session, a battery of standardized exercises [21] was combined with continuous moving visual stimulation in peripheral visual fields lasting the entire duration of the session in order to promote habituation to stimuli with destabilizing effects. The visual stimulation methodology resembled that used during posturography to assess visual hypersensitivity, as described above (Figure 1). The moving scenes changed at each session. In each progressive session, the scenes were the same for all patients.

The battery of standardized exercises included the following: quickly turning the head every 5 s towards the right, middle, and left after vocal command (2 min of duration) while staring at the white line on the display; looking at a point on the display while oscillating the head at 1-hertz frequency on the yaw and pitch plane for 1 min each (Point de Mire); dynamic exercises: all the previous exercises performed while stepping (1 min). Exercises were repeated 4 times with intervals of 5 min.

Before, during, and after each BR-PVS session, VAS-A and VAS-D (continuum from 0, no anxiety/dizziness, to 100, the worst anxiety/dizziness imaginable, on a horizontal line 100 mm in length) [30] were administered.

### 2.4. Statistical Analysis

We provided descriptive statistics as mean, standard deviation (SD), median, interquartile range (IQR), and minimum and maximum values for each variable considered.

We also calculated the percentage of change of each variable after the BR-PVS, as follows: $100 \times (T1 - T0)/T0$; for each calculated change, we provided the same above-mentioned descriptive statistics. For all variables, changes presenting the "minus" sign indicated improvement, except for the Visual-RQ, in which the "minus" sign indicated worsening.

Due to the study design and the very small sample size, we did not provide statistical comparisons according to the available guidelines concerning open-pilot intervention studies [22–24].

The Statistical Package for Windows (Statistica 10.0, Statsoft Inc., Tulsa, OK, USA) was used.

## 3. Results

According to the Consort Transparent Reporting of Trials [23,36], the flow diagram of enrollment is presented in Figure 2.

Eight patients (three men and five women) provided their informed consent to participate. No eligible patient declined to participate. One patient was subsequently excluded because her second Visual-RQ just before the first rehabilitation session was 0.77, while another dropped out after the first rehabilitation session due to unexpected family problems.

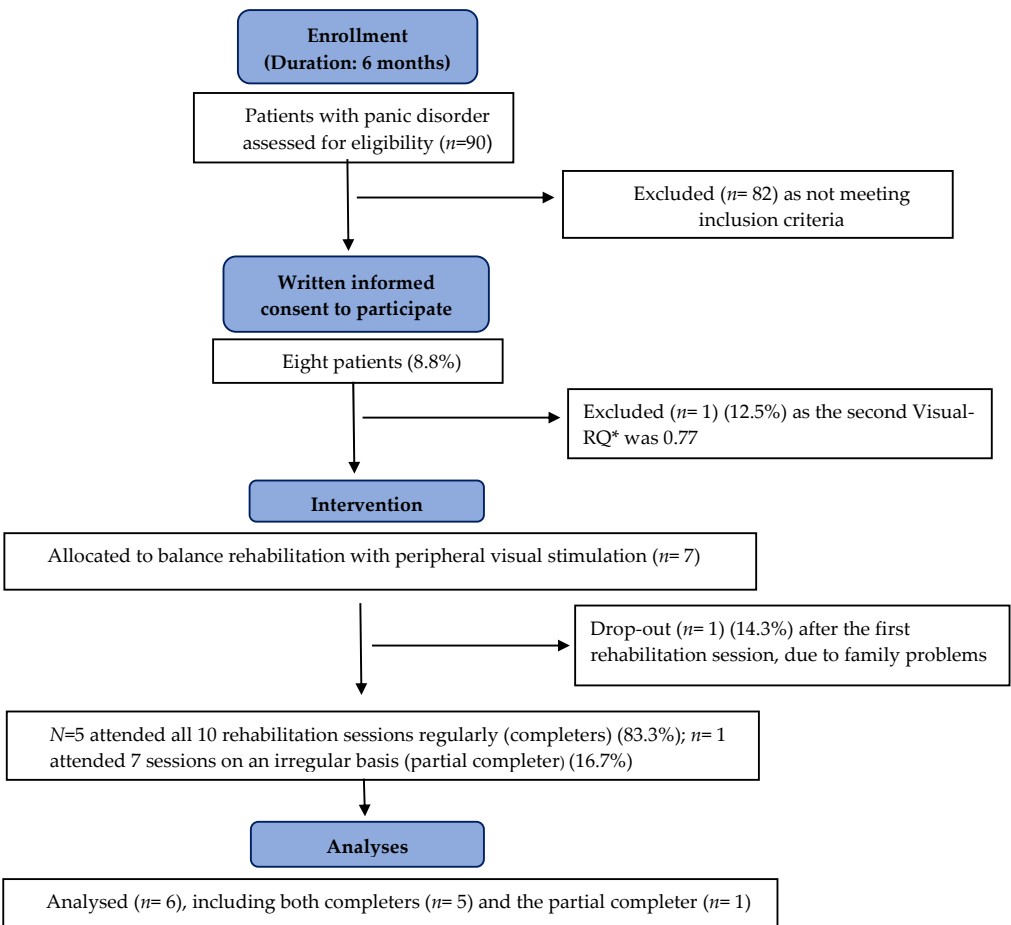

**Figure 2.** Consort flow diagram. * Visual-Romberg Quotient during the second SP-PVS just before the first rehabilitation session.

Our final sample included six patients (three men and three women) (mean (SD) age, 44 (6) years; median (IQR) age, 44 (9); minimum and maximum age, 38 and 55 years, respectively). Three patients had 13 years of education, and three had 8 years of education.

All patients were taking selective serotonin reuptake inhibitors (SSRIs) as follows: two patients, 40 mg of citalopram per day (one of them was the partial completer mentioned below); two patients, 40 mg of paroxetine per day; one patient, 30 mg of paroxetine per day; one patient, 100 mg of sertraline per day.

Five patients attended all 10 rehabilitation sessions regularly (completers), whereas one patient attended only 7 sessions on an irregular basis (partial completer) due to his numerous work commitments.

*3.1. Balance System Function Evaluation*

Three patients presented a persistent non-paroxysmal bipositional apogeotropic nystagmus as a unique vestibular symptom, which was considered a sign of a central vestibular disorder and compatible with their history of probable vestibular migraine [37]. Video HIT, as well as the remaining bedside examination (Head Shaking, skull vibration test, Unterberger), were normal in all patients. No patients reported previous attacks of vertigo during their lifetime.

As required as inclusion criteria, the Visual-RQ of all patients was <0.77 both at T0 (Table 1) and during the second SP-PVS just before the first rehabilitation session (data not shown).

**Table 1.** Clinical characteristics of the sample before and after balance rehabilitation with peripheral visual stimulation. Patients with PD-AG, *n* = 6.

| Characteristics | T0 (Completer, *n* = 5; Partial Completer (PC), *n* = 1) | | | | | | | T1 (Completer, *n* = 5; Partial Completer (PC), *n* = 1) | | | | | | | 100 × (T1 − T0)/T0 (Completer, *n* = 5; Partial Completer (PC), *n* = 1) | | | | | | |
|---|---|---|---|---|---|---|---|---|---|---|---|---|---|---|---|---|---|---|---|---|---|
| | Mean | SD | Median | IQR | Min Value | Max Value | Value of the PC | Mean | SD | Median | IQR | Min Value | Max Value | Value of the PC | Mean | SD | Median | IQR | Min Change | Max Change | Value of the PC |
| Visual-RQ | 0.70 | 0.07 | 0.71 | 0.03 | 0.58 | 0.76 | 0.76 | 0.96 | 0.18 | 1.00 | 0.21 | 0.76 | 1.21 | 0.59 | 37.9 | 21.4 | 39.7 | 14.1 | 7.0 | 65.8 | −22.4 |
| VAS-A before SP-PVS | 37.4 | 20.5 | 46.0 | 26.0 | 7 | 56 | 45 | 16.8 | 15.6 | 24.0 | 28.2 | 0 | 32 | 33 | −67.2 | 30.5 | −53.8 | 56.7 | −39 | −100 | −27 |
| VAS-A during SP-PVS | 46.4 | 31.0 | 51.0 | 45.0 | 12 | 86 | 75 | 17.6 | 20.3 | 12.0 | 29.0 | 0 | 47 | 54 | −75.3 | 25.2 | −76.5 | 45.3 | −45 | −100 | −28 |
| VAS-A after SP-PVS | 44.2 | 31.3 | 50.0 | 38.0 | 6 | 85 | 55 | 18.4 | 20.2 | 17.0 | 27.0 | 0 | 48 | 21 | −72.8 | 26.1 | −66.0 | 45.8 | −44 | −100 | −62 |
| VAS-D before SP-PVS | 40.8 | 26.0 | 45 | 40.0 | 12 | 73 | 47 | 22.0 | 23.2 | 23.0 | 32.0 | 0 | 55 | 24 | −63.5 | 34.5 | −48.9 | 56.1 | −25 | −100 | −49 |
| VAS-D during SP-PVS | 48.0 | 22.9 | 56 | 36.0 | 22 | 74 | 73 | 14.6 | 14.5 | 15.0 | 28.0 | 0 | 30 | 68 | −77.5 | 21.6 | −73.2 | 40.5 | −55 | −100 | −7 |
| VAS-D after SP-PVS | 46.8 | 25.7 | 57 | 36.0 | 15 | 76 | 71 | 13.8 | 13.5 | 19.0 | 19.0 | 0 | 31 | 63 | −78.9 | 19.5 | −68.9 | 33.3 | −59 | −100 | −11 |
| PASS, total score | 9.0 | 2.8 | 8 | 5.0 | 6 | 12 | 8 | 2.2 | 1.6 | 2 | 1 | 1 | 5 | 7 | −75.5 | 13.6 | −75.0 | 19.0 | −58.3 | −91.7 | −13 |
| PASS, unexpected PAs | 0 | 0 | 0 | 0 | 0 | 0 | 0 | 0 | 0 | 0 | 0 | 0 | 0 | 0 | - | - | - | - | - | - | - |
| PASS, expected PAs | 2.0 | 2.0 | 2.0 | 4.0 | 0 | 4 | 3 | 0.4 | 0.9 | 0 | 0 | 0 | 2 | 3 | −83.3 | 28.9 | −100.0 | 25.0 | −50.0 | −100.0 | 0 |
| PASS, AA | 2.4 | 0.5 | 2 | 1.0 | 2 | 3 | 2 | 0.6 | 0.5 | 1 | 1.0 | 0 | 1 | 1 | −76.7 | 22.4 | −66.7 | 33.3 | −50 | −100 | −50 |
| PASS, LS-PAs | 1.8 | 0.4 | 2 | 0 | 1 | 2 | 1 | 0.6 | 0.5 | 1 | 1.0 | 0 | 1 | 1 | −70.0 | 27.4 | −50.0 | 50.0 | −50 | −100 | 0 |
| PASS, AG | 2.8 | 0.8 | 3 | 1.0 | 2 | 4 | 2 | 0.6 | 0.5 | 1 | 1.0 | 0 | 1 | 2 | −78.3 | 21.7 | −75.0 | 23.3 | −50 | −100 | 0 |
| MIA-AAC | 2.1 | 0.7 | 1.8 | 1.1 | 1.4 | 2.9 | 1.5 | 1.6 | 0.5 | 1.4 | 0.6 | 1.0 | 2.3 | 1.5 | −21.4 | 23.4 | −16.5 | 28.9 | 7.3 | −51.6 | −3.3 |
| MIA-AAL | 3.3 | 1.1 | 3.7 | 1.3 | 1.6 | 4.4 | 2.4 | 2.1 | 1.0 | 1.77 | 1.61 | 1.1 | 3.4 | 2.2 | −37.2 | 19.0 | −34.0 | 32.6 | −17.1 | −59.5 | −7.6 |
| MIA-D | 56.8 | 30.5 | 72 | 50 | 20 | 86 | 42 | 28.8 | 22.8 | 32.0 | 36.0 | 6 | 58 | 39 | −55.7 | 21.7 | −62.8 | 28.3 | −25.6 | −78.6 | −7 |
| DHI | 64.8 | 24.9 | 74 | 18.0 | 24 | 88 | 48 | 49.6 | 25.5 | 58 | 22 | 8 | 72 | 44 | −29.7 | 21.1 | −21.6 | 8.5 | −15.4 | −66.7 | −8.3 |

| Characteristics | First Session of the BR-PVS (Completer, *n* = 5; Partial Completer (PC), *n* = 1) | | | | | | | Last Session of the BR-PVS (completer, *n* = 5; Partial Completer (PC), *n* = 1) | | | | | | | 100 × (T1 − T0)/T0 (Completer, *n* = 5; Partial Completer (PC), *n* = 1) | | | | | | |
|---|---|---|---|---|---|---|---|---|---|---|---|---|---|---|---|---|---|---|---|---|---|
| | Mean | SD | Median | IQR | Min Value | Max Value | Value of the PC | Mean | SD | Median | IQR | Min Value | Max Value | Value of the PC | Mean | SD | Median | IQR | Min Change | Max Change | Value of the PC |
| VAS-A before | 38.8 | 31.0 | 28.0 | 30.0 | 5 | 85 | 47 | 30.0 | 31.2 | 20 | 11.0 | 5 | 84 | 38 | −22.3 | 24.7 | −13.0 | 45.3 | 0 | −51 | −19 |
| VAS-A during | 52.0 | 24.4 | 60 | 39 | 20 | 75 | 56 | 29.2 | 27.1 | 26 | 43.0 | 0 | 63 | 45 | −43.8 | 37.5 | −21.2 | 48.3 | −16 | −100 | −20 |
| VAS-A after | 40.0 | 20.1 | 42.0 | 8.0 | 10 | 66 | 30 | 22.0 | 30.9 | 0 | 45.0 | 0 | 65 | 27 | −60.3 | 54.4 | −100 | 98.5 | 0 | −100 | −10 |
| VAS-D before | 37.6 | 24.1 | 40 | 28.0 | 5 | 67 | 69 | 29.4 | 23.0 | 23 | 20.0 | 5 | 65 | 57 | −16.4 | 28.6 | −4.2 | 4.5 | 0 | −67 | −17 |
| VAS-D during | 60.0 | 26.3 | 59 | 32.0 | 25 | 92 | 77 | 19.2 | 11.7 | 26 | 10.0 | 0 | 28 | 60 | −71.5 | 16.7 | −66.7 | 4.3 | −56 | −100 | −22 |
| VAS-D after | 44.0 | 21.4 | 50 | 20 | 10 | 64 | 50 | 30.0 | 21.5 | 29 | 27.0 | 10 | 62 | 54 | −25.2 | 35.9 | −3.1 | 42.0 | 0 | −81 | 8 |

AA: anticipatory anxiety; AG: agoraphobia; BR-PVS: balance rehabilitation with peripheral visual stimulation; DHI: Dizziness Handicap Inventory; IQR: interquartile range; LS-PAs: limited symptom panic attacks; MIA-AAC: Mobility Inventory for Agoraphobia-Avoidance Accompanied scale; MIA-AAL: Mobility Inventory for Agoraphobia-Avoidance Alone scale; MIA-D: Mobility Inventory for Agoraphobia-Dizziness; *n*: number; Max = maximum; Min: minimum; PAs: panic attacks; PASS: Panic Associated Symptom Scale; PC: partial completer; PD-AG: Panic disorder with agoraphobia; SD: standard deviation; SP-PVS: Static posturography with moving peripheral visual stimulation; T0: One week before the beginning of the balance rehabilitation sesssions; T1: One week after the end of the balance rehabilitation sesssions; VAS-A: Visual analog scale for anxiety; VAS-D: Visual analog scale for dizziness; Visual-RQ: Visual-Romberg Quotient. The lower the quotient, the greater the postural instability induced by visual stimulation; the Visual-RQ cut-off value to identify the visual hypersensitivity was <0.77.

At T1 (Table 1), the completers presented a general improvement of the Visual-RQ, ranging from 7% to 65.8%. Considering each patient individually, four patients achieved Visual-RQ values >0.77, and one achieved a value of 0.76 starting from 0.71 at T0. Only the partial completer reported a decrease in Visual-RQ value at T1 (0.59), starting from 0.75 at T0 (22.4% worsening).

*3.2. Dizziness and Panic–Agoraphobic Symptom Evaluation*

At T1 (Table 1), the completers exhibited a general improvement in the total DHI score, whose decrease ranged from 15.4% to 66.7 %; the partial completer presented a more limited decrease in DHI total score (8.3%).

Most clinical measures assessing panic–agoraphobic symptoms and dizziness in daily life indicated a considerable improvement at T1 among completers, while the partial completer had only limited improvement (Table 1).

Anxiety and dizziness during the SP-PVS at T1 and the last rehabilitation session globally decreased, even though to a lesser extent in the partial completer (Table 1).

## 4. Discussion

In this open-pilot intervention study, we evaluated initial evidence for feasibility, acceptability, and potential clinical usefulness of 10 sessions of a newly developed BR-PVS in six outpatients with PD-AG who presented residual agoraphobia, even after SSRI pharmacotherapy and CBT, and a balance system-related clinical profile, namely persistent dizziness in daily life and peripheral visual hypersensitivity as identified by SP-PVS.

No patients presented peripheral vestibular abnormalities, while three patients reported signs and symptoms compatible with a history of migraine, in line with the previously reported association between PD and migraine [7].

The intervention was dedicated to patients with PD-AG who exhibited a highly specific clinical profile, and the sample size was very small. However, this pilot study showed reasonable levels of feasibility, acceptability, and potential clinical usefulness of BR-PVS, suggesting this intervention is worth being tested in larger and adequately powered randomized controlled studies.

Approximately 9% of patients with PD visited by a single psychiatrist in 6 months during a limited amount of monthly hours dedicated to outpatient visits were eligible. Even considering the rates of patients excluded or dropping out for different reasons after written informed consent, it seems plausible that future enrollments conducted by larger psychiatric personnel in specialized centers for treating anxiety Disorders may reach adequate sample sizes to calculate statistical significance and effect sizes. Similarly, larger personnel dedicated to the study may partly decrease potential partial completers due to personal, familial, or work commitments. Offering the opportunity to choose between different time slots during which BR-PVS can take place may increase the chance of completing the relatively high number of BR-PVS sessions. Furthermore, future multicenter studies involving different specialized centers may contribute to enlarging sample sizes and increasing the generalizability of the results.

A limitation of our study is the lack of questionnaires to assess participants' perceptions, opinions, and satisfaction related to the BR-PVS. However, the fact that no eligible patients declined to participate and no patients dropped out due to intervention-related issues may be indirect preliminary indications of reasonable acceptability. Future studies should address this limitation by using validated tools to evaluate BR-PVS acceptability directly.

Finally, instrumentations and set-up were relatively easy to implement and had limited costs, increasing the replicability and facilitating future BR-PVS-related studies in different centers.

Although the sample size was not suitable for statistical comparisons, the observation of both posturographic and clinical measures revealed improvement after the intervention, suggesting the potential clinical usefulness of BR-PVS in these patients.

Four patients achieved normalization of postural control during posturography-PVS, as indicated by Visual-RQ values > 0.77. One patient showed consistent improvement relative to her Visual-RQ before rehabilitation, achieving a value of 0.76. Although this value did not indicate complete normalization, it is conceivable that additional sessions could have promoted further amelioration. Only the partial completer reported a decrease in his Visual-RQ after rehabilitation, possibly related to the lower number of attended sessions and discontinuous attendance.

In line with the improvement of posturographic measures, anxiety and dizziness perceived during both the last rehabilitation session and posturography decreased, even though to a lesser extent in the partial completer.

After BR-PVS, patients exhibited a global decrease in panic–agoraphobic symptoms, dizziness perceived in agoraphobia-related situations, and self-perceived handicapping effects caused by dizziness in daily life, as measured by the PASS-total score, PASS-anticipatory anxiety, PASS-agoraphobia, and expected PAs in agoraphobic situations, MIA-AAL, MIA-D, and DHI, respectively. Only the partial completer showed minimal clinical improvement.

Notably, residual pre-BR-PVS panic-agoraphobic symptoms of participants encompassed a variety of agoraphobia-related symptoms, whereas no patients presented unexpected PAs. This picture was coherent with the expected remission of unexpected PAs achieved with antipanic medications, while the visual hypersensitivity may have contributed to the persistence of agoraphobia despite treatment. It is conceivable that small postural modifications or stimuli from complex sensorial environments (e.g., shopping malls, traffic, and crowds) may provoke dizziness, instability, and discomfort in patients with panic disorder and visual hypersensitivity, thus contributing to triggering situational anxiety or panic. Defensive avoidance mechanisms induced by repeated signals of instability, interoceptive and exteroceptive conditioning processes related to destabilizing stimuli, and operant learning processes related to the avoidance of disturbing environments in everyday life may influence the development, severity, or maintenance of agoraphobia in these patients [18,19,38,39]. Consistently, our results suggest that the decrease in peripheral visual hypersensitivity during posturography after rehabilitation may be associated with the improvement of agoraphobia and dizziness in daily life, possibly via sensory re-integration processes and habituation mechanisms that promote physical and emotional desensitization to complex sensorial environments.

However, the study design does not allow us to conclude that the clinical improvement was entirely attributable to the specific effects of BR-PVS. Although participants had previously completed an adequate CBT program, we cannot exclude those additional CBT sessions that would have provided benefits on the residual symptoms. Likewise, we cannot rule out that other kinds of rehabilitation, not including visual stimulation, would have led to similar improvements. For these reasons, future randomized studies with active comparators in larger samples are warranted to understand whether our preliminary suggestions can be confirmed and generalized.

This study did not include follow-up monitoring, and further studies should evaluate the long-term stability of improvements obtained during BR-PVS.

We applied BR-PVS using conventional instruments. However, it is conceivable that future applications of virtual reality-based rehabilitation may provide greater benefits by offering experiences that are more immersive and more closely resemble daily life environments, as demonstrated in other clinical conditions [40–42].

## 5. Conclusions

In conclusion, this open-pilot intervention study showed reasonable levels of feasibility and acceptability, and potential clinical usefulness, of BR-PVS in patients with PD-AGO, residual agoraphobia, and a balance system-related clinical profile. Overall, this study suggests BR-PVS as a potential therapeutic option worth being tested in larger and adequately powered randomized controlled studies. Within the framework of personalized

psychiatry and a multidisciplinary approach to psychiatric disorders [43], balance system evaluation should be considered in clinical practice in patients with residual agoraphobia, especially in case of complaints about dizziness or subjective discomfort in complex visual environments. If confirmed in adequate and reliable studies, BR-PVS might be a future adjunctive therapeutic option for these highly suffering patients.

**Author Contributions:** Conceptualization and methodology, D.C., C.C., G.P. and R.T.; investigation, D.C., C.C. and R.T.; formal analysis and visualization, D.C., S.D., M.G., G.P. and R.T.; writing—original draft, D.C. and R.T.; writing—review and editing, D.C., C.C., S.D., M.G., G.P. and R.T.; supervision, D.C. and R.T. All authors made substantive intellectual contributions to the manuscript, revised it, and provided substantial comments. All authors have read and agreed to the published version of the manuscript.

**Funding:** This study received no external funding.

**Institutional Review Board Statement:** This study adheres to the principles of the Helsinki Declaration and was approved in 2011 by the ASL of Milan Ethics Committee ASL "City of Milan".

**Informed Consent Statement:** Written informed consent was obtained from all participants involved in the study.

**Data Availability Statement:** The data supporting the presented results of the pilot study are available on request from the corresponding author. The data are not publicly available to protect the privacy of the participants.

**Conflicts of Interest:** The authors declare no conflict of interest.

## Abbreviations

| | |
|---|---|
| AA | anticipatory anxiety |
| AG | agoraphobia |
| BR-PVS | balance rehabilitation with peripheral visual stimulation |
| CBT | cognitive–behavioral therapy |
| IQR | interquartile range |
| LS-PAs | limited symptom panic attacks |
| MIA | mobility inventory for agoraphobia |
| MIA-D | mobility inventory for agoraphobia, modified for dizziness |
| PA(s) | panic attacks |
| PASS | panic-associated symptom scale |
| PD-AG | panic disorder with agoraphobia |
| PVS | peripheral visual stimulation |
| RQ | Romberg Quotient |
| SD | standard deviation |
| SP | static posturography |
| SP-PVS | static posturography with peripheral visual stimulation |
| SSRI(s) | selective serotonin reuptake inhibitor(s) |
| VAS-A | visual analog scale for anxiety |
| VAS-D | visual analog scale for dizziness |

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
