# Peer review of "Balance Rehabilitation with Peripheral Visual Stimulation in Patients with Panic Disorder and Agoraphobia: An Open-Pilot Intervention Study"

_audiolres, doi:10.3390/audiolres13030027_

Round 1
Reviewer 1 Report
Caldirola et al provide a pilot study short report on vestibular rehabilitation to alleviate panic symptoms among a dizzy subgroup of subjects with panic disorder. The idea of overlapping issues between dizziness and panic disorders is interesting and the treatment strategy holds potential, if shown effective in future larger studies with a control group.
Abstract: Did the participants have audiovestibular diagnoses outside the panic disorder spectrum? How were the participants treated before the intervention (“…presented with residual panic-phobic symptoms after treatment…”)
Introduction: The authors describe an association between PD-AG and balance disorders in terms which are reminiscent of 3PV (https://pubmed.ncbi.nlm.nih.gov/29208729/). Is it possible that these conditions are parts of the same spectrum of diseases? Perhaps mentioning 3PV as an alternative cause of dizziness in the participants would be prudent?
Discussion: The authors state that both posturographic and clinical measures revealed considerable improvement after rehabilitation (row 248). It is not recommendable to use non-significant findings (VRQ) in such strong statements in the discussion section. This was a pilot study with very few participants and the reader do not expect great power to show differences between T0 and T1.
Author Response
Reviewer 1
Caldirola et al provide a pilot study short report on vestibular rehabilitation to alleviate panic symptoms among a dizzy subgroup of subjects with panic disorder. The idea of overlapping issues between dizziness and panic disorders is interesting and the treatment strategy holds potential, if shown effective in future larger studies with a control group.
Dear Reviewer, we thank you for the attention dedicated to our manuscript and the useful suggestions. All the modifications made in the text are red coloured.
Abstract: Did the participants have audiovestibular diagnoses outside the panic disorder spectrum? How were the participants treated before the intervention (“…presented with residual panic-phobic symptoms after treatment…”)
According to your suggestion, in the Abstract we specified that no patients presented peripheral vestibular abnormalities, and that patients presented residual agoraphobia after SSRI treatment and cognitive-behavioral therapy.
Introduction: The authors describe an association between PD-AG and balance disorders in terms which are reminiscent of 3PV (https://pubmed.ncbi.nlm.nih.gov/29208729/). Is it possible that these conditions are parts of the same spectrum of diseases? Perhaps mentioning 3PV as an alternative cause of dizziness in the participants would be prudent?
We agree with your comment. In the Introduction we mentioned this open issue as follows:
“Finally, dizziness and unsteadiness in patients with PD-G may belong to the spectrum of persistent postural-perceptual dizziness (PPPD), possibly sharing some common mechanisms [16,17].
We also included the reference you suggested (16) and an additional one (17):
- Popkirov, S.; Staab, J.P.; Stone, J. Persistent Postural-Perceptual Dizziness (PPPD): A Common, Characteristic and Treatable Cause of Chronic Dizziness. Pract. Neurol. 2018, 18, 5–13, doi:10.1136/PRACTNEUROL-2017-001809.
- Waterston, J.; Chen, L.; Mahony, K.; Gencarelli, J.; Stuart, G. Persistent Postural-Perceptual Dizziness: Precipitating Conditions, Co-Morbidities and Treatment With Cognitive Behavioral Therapy. Front. Neurol. 2021, 12, doi:10.3389/FNEUR.2021.795516.
Discussion: The authors state that both posturographic and clinical measures revealed considerable improvement after rehabilitation (row 248). It is not recommendable to use non-significant findings (VRQ) in such strong statements in the discussion section. This was a pilot study with very few participants and the reader do not expect great power to show differences between T0 and T1.
Thank you for your comment. Based on your suggestions and those from Reviewer 2, we amended the manuscript (description of study details, results, and discussion) following the current guidelines and recommendations for reporting non-randomized pilot and feasibility studies [22–24].
- Lancaster, G.A.; Thabane, L. Guidelines for Reporting Non-Randomised Pilot and Feasibility Studies. Pilot feasibility Stud. 2019, 5, doi:10.1186/S40814-019-0499-1.
- Hoffmann, T.C.; Glasziou, P.P.; Boutron, I.; Milne, R.; Perera, R.; Moher, D.; Altman, D.G.; Barbour, V.; Macdonald, H.; Johnston, M.; et al. Better Reporting of Interventions: Template for Intervention Description and Replication (TIDieR) Checklist and Guide. BMJ 2014, 348, doi:10.1136/BMJ.G1687.
- Chan, C.L. A Website for Pilot and Feasibility Studies: Giving Your Research the Best Chance of Success. Pilot feasibility Stud. 2019, 5, doi:10.1186/S40814-019-0522-6.

Reviewer 2 Report
I enjoyed reading this study of vestibular rehabilitation with peripheral visual stimulation for people with panic disorder with agoraphobia. The authors provide sound justification for their study by citing the evidence of balance system abnormalities in the pathophysiology of panic disorder and the frequency of dizziness and balance problems in this patient group, in addition to the lack of previous research that has sought to apply vestibular rehabilitation for such patients.
The pilot study included 6 participants undergoing 10 sessions of balance rehabilitation over 3 weeks. It would be helpful for the authors to provide a flow diagram of recruitment to determine what percentage of patients approached to participate eventually consented to be included and how many people were excluded and for what reasons. The authors point out that they used a very selected group of patients and 6 patients over 8 months would seem very low and possibly not representative, and could suggest that a future trial is not feasible. It is crucial to accurately identify the recruitment rate through a pilot study.
Likewise, I think I understood the therapy set-up with the video screens but it would be helpful to have a picture of this set-up for clarity and replication.
My main issue with the paper is the focus on statistical outcome rather than pilot study outcomes. A pilot study asks whether something can be done, should the researchers proceed with it, and if so, how. It can also examine recruitment potential and other feasibility measures etc. It can provide some evidence as to whether a treatment may be effective but it is inappropriate to conduct and report on statistical tests for a pilot study with such low sample size. Instead, discussion should be focused on whether a main trial would be feasible and the data should be focused on descriptive data with means/medians/ranges/IQR/confidence intervals etc. Throughout the results the authors should provide the data related to variability (ranges/IQR) rather than just the median.
Furthermore, since pilot studies are not for testing clinical effectiveness/hypothesis testing they should change the aim of the study to be in line with a pilot study.
I recommend that the authors familiarize themselves with reporting standards for pilot studies and amend the manuscript accordingly.
Author Response
Reviewer 2
I enjoyed reading this study of vestibular rehabilitation with peripheral visual stimulation for people with panic disorder with agoraphobia. The authors provide sound justification for their study by citing the evidence of balance system abnormalities in the pathophysiology of panic disorder and the frequency of dizziness and balance problems in this patient group, in addition to the lack of previous research that has sought to apply vestibular rehabilitation for such patients.
Dear Reviewer, we sincerely thank you for the attention dedicated to our manuscript and very useful suggestions that helped us improve the quality of the manuscript.
All the modifications made in the text are red coloured.
As you recommended, we familiarized with reporting standards for pilot studies and amended the manuscript accordingly. Hence, we modified title (i.e., “Balance rehabilitation with peripheral visual stimulation in patients with panic disorder and agoraphobia: an open-pilot intervention study”), abstract, and the main text, providing appropriate aims, reporting, and related discussion. We removed any statistical comparisons (not appropriate for aims and sample size) and provided full reporting of descriptive statistics.
At the end of the “Introduction” we clearly stated that…
”Throughout the manuscript, reporting and related discussion follow the current guidelines and recommendations for reporting non-randomized pilot and feasibility studies [22–24].”
- Lancaster, G.A.; Thabane, L. Guidelines for Reporting Non-Randomised Pilot and Feasibility Studies. Pilot feasibility Stud. 2019, 5, doi:10.1186/S40814-019-0499-1.
- Hoffmann, T.C.; Glasziou, P.P.; Boutron, I.; Milne, R.; Perera, R.; Moher, D.; Altman, D.G.; Barbour, V.; Macdonald, H.; Johnston, M.; et al. Better Reporting of Interventions: Template for Intervention Description and Replication (TIDieR) Checklist and Guide. BMJ 2014, 348, doi:10.1136/BMJ.G1687.
- Chan, C.L. A Website for Pilot and Feasibility Studies: Giving Your Research the Best Chance of Success. Pilot feasibility Stud. 2019, 5, doi:10.1186/S40814-019-0522-6.
The pilot study included 6 participants undergoing 10 sessions of balance rehabilitation over 3 weeks. It would be helpful for the authors to provide a flow diagram of recruitment to determine what percentage of patients approached to participate eventually consented to be included and how many people were excluded and for what reasons. The authors point out that they used a very selected group of patients and 6 patients over 8 months would seem very low and possibly not representative, and could suggest that a future trial is not feasible. It is crucial to accurately identify the recruitment rate through a pilot study.
According to your suggestions, in the section “Results” we added Figure 2 to show the recruitment flow diagram, according to the Consort Transparent Reporting of Trials [23,36].
- Hoffmann, T.C.; Glasziou, P.P.; Boutron, I.; Milne, R.; Perera, R.; Moher, D.; Altman, D.G.; Barbour, V.; Macdonald, H.; Johnston, M.; et al. Better Reporting of Interventions: Template for Intervention Description and Replication (TIDieR) Checklist and Guide. BMJ 2014, 348, doi:10.1136/BMJ.G1687.
- Xie, E.B.; Freeman, M.; Penner-Goeke, L.; Reynolds, K.; Lebel, C.; Giesbrecht, G.F.; Rioux, C.; MacKinnon, A.; Sauer-Zavala, S.; Roos, L.E.; et al. Building Emotional Awareness and Mental Health (BEAM): An Open-Pilot and Feasibility Study of a Digital Mental Health and Parenting Intervention for Mothers of Infants. Pilot feasibility Stud. 2023, 9, doi:10.1186/S40814-023-01245-X.
In addition, throughout the section “Materials and Methods” we added as much information as possible relevant to evaluate levels of feasibility and replicability to inform possible future larger randomized controlled studies. In the section “Discussion” we commented on these issues
Likewise, I think I understood the therapy set-up with the video screens but it would be helpful to have a picture of this set-up for clarity and replication.
We included Figure 1 to show the therapy set-up with the video screens
My main issue with the paper is the focus on statistical outcome rather than pilot study outcomes. A pilot study asks whether something can be done, should the researchers proceed with it, and if so, how. It can also examine recruitment potential and other feasibility measures etc. It can provide some evidence as to whether a treatment may be effective but it is inappropriate to conduct and report on statistical tests for a pilot study with such low sample size. Instead, discussion should be focused on whether a main trial would be feasible and the data should be focused on descriptive data with means/medians/ranges/IQR/confidence intervals etc. Throughout the results the authors should provide the data related to variability (ranges/IQR) rather than just the median.
All done. As described above, we changed the aims and, accordingly, we amended the results and discussion. We removed any statistical comparisons and provided in Table 1 appropriate descriptive statistics. Related comments were provided in the “Discussion”.
Furthermore, since pilot studies are not for testing clinical effectiveness/hypothesis testing they should change the aim of the study to be in line with a pilot study.
All done, as described above.
I recommend that the authors familiarize themselves with reporting standards for pilot studies and amend the manuscript accordingly.
All done, as described above.
Round 2
Reviewer 2 Report
The authors have submitted a vastly improved manuscript. I have no further suggestions.